# Temperature Differentially Affects Gene Expression in Antarctic Thraustochytrid *Oblongichytrium* sp. RT2316-13

**DOI:** 10.3390/md18110563

**Published:** 2020-11-18

**Authors:** Paris Paredes, Giovanni Larama, Liset Flores, Allison Leyton, Carmen Gloria Ili, Juan A. Asenjo, Yusuf Chisti, Carolina Shene

**Affiliations:** 1Department of Chemical Engineering, Center of Food Biotechnology and Bioseparations, BIOREN, and Centre of Biotechnology and Bioengineering (CeBiB), Universidad de La Frontera, Av. Francisco Salazar 01145, Temuco 4780000, Chile; p.paredes03@ufromail.cl (P.P.); liset.flores@ufrontera.cl (L.F.); allison.leyton@ufrontera.cl (A.L.); 2Centro de Modelación y Computación Científica, Universidad de La Frontera, Av. Francisco Salazar 01145, Temuco 4780000, Chile; giovanni.larama@ufrontera.cl; 3Centro de Excelencia en Medicina Traslacional—Scientific and Technological Bioresource Nucleus (CEMT-BIOREN), Universidad de La Frontera, Av. Alemania 0478, Temuco 4810296, Chile; carmen.ili@ufrontera.cl; 4Centre for Biotechnology and Bioengineering (CeBiB), Department of Chemical Engineering and Biotechnology, Universidad de Chile, Beauchef 851, Santiago 8370459, Chile; juasenjo@ing.uchile.cl; 5School of Engineering, Massey University, Private Bag 11 222, Palmerston North 4442, New Zealand; Y.Chisti@massey.ac.nz

**Keywords:** gene expression analysis, lipid metabolism, *Oblongichytrium* sp., polyunsaturated fatty acid synthesis, thraustochytrids

## Abstract

*Oblongichytrium* RT2316-13 synthesizes lipids rich in eicosapentaenoic acid (EPA) and docosahexaenoic acid (DHA). The content of these fatty acids in the total lipids depended on growth temperature. Sequencing technology was used in this work to examine the thraustochytrid’s response to a decrease in growth temperature from 15 °C to 5 °C. Around 4% (2944) of the genes were differentially expressed (DE) and only a few of the DE genes (533 upregulated; 206 downregulated) had significant matches to those in the SwissProt database. Most of the annotated DE genes were related to cell membrane composition (fatty acids, sterols, phosphatidylinositol), the membrane enzymes linked to cell energetics, and membrane structure (cytoskeletal proteins and enzymes). In RT2316-13, the synthesis of long-chain polyunsaturated fatty acids occurred through ω3- and ω6-pathways. Enzymes of the alternative pathways (Δ8-desaturase and Δ9-elongase) were also expressed. The upregulation of the genes coding for a Δ5-desaturase and a Δ5-elongase involved in the synthesis of EPA and DHA, explained the enrichment of total lipid with these two long-chain fatty acids at the low temperature. This molecular response has the potential to be used for producing microbial lipids with a fatty acids profile similar to that of fish oils.

## 1. Introduction

Marine thraustochytrids (heterotrophic biflagellate single-celled eukaryotes) are good producers of lipids rich in docosahexaenoic acid (DHA, C22:6^Δ4,7,10,13,16,19^). In fact, commercial production of DHA for human food relies on thraustochytrids such as *Schizochytrium* sp. and *Ulkenia* sp. [1,2]. Although most of the studied thraustochytrids produce DHA as the main long chain polyunsaturated fatty acid (LCPUFA), other strains in this group also synthesize eicosapentaenoic acid (EPA, C20:5^Δ5,8,11,14,17^). Dietary consumption of lipids rich in EPA and DHA has many beneficial effects on health [3]. At present, marine fish are the main source of oils rich in EPA and DHA. Many microorganisms synthesize EPA and DHA and may in the future contribute to biotechnological production of lipids similar to fish oil.

Cold-adapted microbes commonly have an elevated level of unsaturated fatty acids in their cell membranes [4,5,6] possibly because lipid bilayers containing unsaturated fatty acids retain fluidity at relatively low temperatures [7] compared to bilayers with a preponderance of saturated fatty acids. Elevated levels of DHA and/or EPA have been observed in some marine species grown at low temperature [8,9,10], but this is not always so [11] and there is some uncertainty about the exact influence of EPA and DHA on the properties of lipid bilayer membranes. For example, although a reduced culture temperature elevated the EPA level, this fatty acid was claimed to not influence the bulk bilayer fluidity in the gram-negative bacterium *Shewanella livingstonensis* Ac10 [6]. Similarly, in the gram-negative deep-sea bacterium *Photobacterium profundum* SS9, EPA levels did increase in low temperature cultivation, but this was concluded to be nonessential for low-temperature growth [4]. Levels of EPA and DHA in thraustochytrid total lipids depend on the culture conditions [12,13], including temperature of cultivation [14].

The thraustochytrid *Oblongichytrium* RT2316-13 isolated from Antarctic seawaters [15] was the focus of the present study. This microorganism grew at temperatures ranging from 2 to 15 °C and, therefore, could be classified as a psychrophile [16]. RT2316-13 was shown to produce DHA and EPA [14]. Cultures grown at 5 and 15 °C attained the same final biomass concentration, although at 15 °C the specific growth rate was 4.6-fold greater than at 5 °C (0.032 h^−1^) [14]. Although the biomass grown at 5 °C had significantly fewer lipids than the biomass grown at 15 °C, the proportions of DHA and EPA in the total lipids were higher in the cells grown at the lower temperature [14]. Therefore, the growth temperature was postulated to differentially affect the expression of at least some of the genes relating to the metabolism of fatty acids. To better understand the temperature-dependent metabolism of LCPUFA in *Oblongichytrium* RT2316-13, the present work examined the differential expression of genes by the microorganism subjected to a low temperature stress (5 °C) in comparison with control (15 °C). The genes relating to lipid metabolism, especially the synthesis of LCPUFA, were focused on.

## 2. Results and Discussion

### 2.1. Sequence Analysis and Transcriptome Annotation

The total output of HiSeq 4000 for all the samples was approximately 174 million paired-end reads, with an average of 29 million reads for each sample. The quality filtering step retained over 99.5% of the reads (173.7 million), confirming an excellent sequencing performance and assuring a minimum of 3 Gbps of data (20 million reads) per sample (Appendix A). The final transcriptome resulted in 118,595 transcript sequences that were contained in 69,220 genes. The quality metrics indicated that 50% of the assembled bases were contained in transcripts longer than 2518 bp (N50), with an average transcript length of 1243 bp, and a total size of 147 Mbp. The presence of several short transcripts in the assembly was evidenced by a median length metric of 554 bp (half of the assembled transcripts had a length of 554 bp or less) and this may have affected the average length of the transcriptome (Table 1). The results from BUSCO (Benchmarking Universal Single-Copy Orthologs [17]) indicated a high degree of transcriptome completeness, resulting in 235 of the 255 genes used for evaluation being found (Table 2).

Alignment of high quality reads to the assembled transcriptome of RT2316-13 showed a high mapping rate: an average of 91.63% for the properly aligned paired reads, and an overall mapping rate > 99% for every sample. The overall alignment included reads that contained one mismatch in their sequences, and reads that were aligned separately (Appendix A).

### 2.2. Differentially Expressed Genes

Using a false discovery rate (FDR) of 0.05 and a fold change (FC) of 4 as thresholds, 2944 genes were found to be differentially expressed (DE). Of these DE genes, 2062 were induced by the low temperature (LT) whereas 882 genes were repressed (Excel file Paredes_et_al.xlsx, Appendix A). The homology searches in the public database SwissProt indicated that relatively few of the DE genes (25.1%) had significant matches, probably because of a comparative rarity of microbes of the genus *Oblongichytrium*. Of the annotated genes (1999 transcripts), 533 were induced and 206 were repressed (Excel file Paredes_et_al.xlsx, Appendix A). Most of the DE genes belonged to the GO-slim (cut-down versions of the Gene Ontologies) cellular component category (Figure 1a). The DE proteins were classified into 16 classes (Figure 1b). Among the induced proteins, the class “Nucleic acid binding” (PC00031, *p* = 0.0187) was depleted, while “Cytoskeletal protein” (PC00085, *p* = 0.0006) was overrepresented. In the repressed protein group the overrepresented classes were the “Translation protein” (PC00263, *p* < 0.0001) and its subclass “Ribosomal protein” (PC00202, *p* < 0.0001). The subclass “Oxidoreduction”, whose parent was “Metabolite interconversion enzyme”, was also overrepresented (PC00176, *p* = 0.0022). Genes relating to carbohydrate, amino acids, energy, and lipid metabolism represented 64.4% of the annotated genes (Figure 1c). In the lipid metabolism group, the genes relating to fatty acid degradation, and metabolism of glycerophospholipid and glycerolipid, accounted for 44.5% of the annotated genes whereas 6.2% of the annotated genes related to fatty acid elongation (Figure 1d). Annotated genes related to carbohydrate metabolism (21%, Figure 1e) were the most abundant and within this group more than 52% corresponded to genes coding for enzymes related to glycolysis, inositol phosphate, citrate cycle, and pyruvate metabolisms.

#### 2.2.1. Cytoskeletal Proteins

Forty-eight genes coding for the cytoskeleton-related proteins were DE. The cytoskeleton of eukaryotes is a dynamic network of interlinking protein filaments that confer shape and mechanical resistance to cells, and allows them to move. Cytoskeleton is also involved in many cell signaling pathways and intracellular transport. The cytoskeleton is composed of microfilaments (linear polymers of G-actin proteins), intermediate filaments, and microtubules (MT, polymers of tubulin). In RT2316-13, the genes coding for tubulin α-chain (2 genes) (FC = 10.88; 16.46), tubulin tyrosine ligase (FC = 11.73), tubulin polymerization-promoting protein family member (FC = 17.02), and actin (2 genes) (FC = 8.47; 27.03) were upregulated.

Cilia are evolutionarily conserved, MT-based organelles that play important roles in cell movement, environment sensing, and signal transduction. Motile biflagellate zoospores are characteristic of most thraustochytrids [18], including RT2316-13 (microscopic observation). Eighteen genes coding for cilia and flagella-associated proteins were upregulated (FC in the range of 5.34 to 18.25). The assembly, maintenance, and functions of cilia depend on the intraflagellar transport (IFT), the bidirectional trafficking of vesicles and proteins along the ciliary axoneme [19]. IFT is mediated by motor protein complexes, the cytoplasmic dyneins, and kinesins. In RT2313-13, the expression of genes coding for kinesin-related proteins (19 genes) (FC in the range of 4.41 to 15.84), and dynein-related proteins (24 genes) (FC in the range of 4.49 to 19.96) were upregulated.

The contribution of cytoskeleton to the cell’s structural integrity, organization, and movement explained the significant number of the DE genes in RT2316-13 in response to a change in growth temperature. Most of the annotated genes were upregulated which suggested that the cells either needed to adapt or reshape their structure, or elevate the level of certain intracellular enzymes, to enable movement at the low temperature. 

#### 2.2.2. Inositol Phosphate Metabolism

Phosphatidylinositol (PI) is a family of lipids that are plentiful in cell membranes [20]. In one of its roles, PI modulates protein function, depending on the phosphorylation state of inositol. Deletion of INP51 (a gene coding a protein with 5-phosphatase activity against phosphatidylinositol 4,5-bisphosphate, PI(4,5)P2) in *Saccharomyces cerevisiae* resulted in accumulation of PI(4,5)P2 that correlated to cold-tolerance through an unknown mechanism [21]. In RT2316-13, the genes encoding phosphatidylinositol 4-phosphate 5-kinase (FC = 4.79; 7.92; 18.75) and phosphatidylinositol 4-kinase (FC = 8.19), and the enzymes involved in PI(4,5)P2 synthesis, were upregulated, suggesting a role in cold-tolerance as previously observed in *S. cerevisiae*. 

PI(4,5)P2 is hydrolyzed by phospholipase C (PLC), an enzyme known to activate at low temperatures [22,23], to produce the secondary messengers diacylglycerol (DAG) and inositol-1,4,5-trisphosphate. In RT2316-13 a gene coding for a PLC (FC = 13.77) was upregulated. Once formed, DAG activated protein kinase C (PKC), an enzyme that has a central role in the cell wall integrity signaling pathway that responds to cell surface stress in *S. cerevisiae* [24]. In RT2316-13 a gene coding for a Ca^2+^-independent, phospholipid- and DAG-dependent serine/threonine-protein kinase (novel PKC) was upregulated (FC = 16.67). Upregulation of the genes relating to the synthesis of PI(4,5)P2, PLC and PKC suggested LT as a stress factor for RT2316-13. 

#### 2.2.3. Glycolysis, Pentose Phosphate Pathway, and Tricarboxylic Acid Cycle

Four genes coding for enzymes in the glycolysis pathway were upregulated by LT in RT2316-13. These genes related to phosphoglycerate kinase (FC = 14.09), enolase (FC = 8.81), phosphoglycerate mutase (FC = 5.70), and phosphoenolpyruvate synthase (FC = 6.24). Upregulation of the latter gene might enable cells to use three-carbon substrates (e.g., lactate, pyruvate, and alanine) as carbon sources [25]. The LT treatment upregulated a gene coding for a probable 6-phosphofructo-2-kinase (FC = 7.12) in RT2316-13. This enzyme catalyzes the production of fructose-2,6-bisphosphate which allosterically activates 6-phosphofructo-1-kinase, a rate-limiting enzyme and an essential control point in the glycolytic pathway. Genes coding for 2,3-bisphosphoglycerate-dependent phosphoglycerate mutase (FC = −9.44), fructose-bisphosphate aldolase (FC = −17.70), a NADP-dependent alcohol dehydrogenase (FC = −17.32), and acetyl-coenzyme A synthetase (FC = −16.51), were downregulated. 

Four genes coding for enzymes in the pentose phosphate pathway were DE in RT2316-13. Among these, a gene coding for NADPH-producing 6-phosphogluconate dehydrogenase, a source of NADPH for fatty acid biosynthesis, was downregulated (FC = −17.01). On the other hand, genes coding for a ribose-phosphate pyrophosphokinase 4 (FC = 4.30), involved in nucleotide biosynthesis, and ribokinase (FC = 11.30) which catalyzes the phosphorylation of ribose to ribose-5-phosphate, were upregulated. The upregulation of the ribokinase gene could be related to a possible degradation of ribosomes to release nutrients [26]. The LT treatment downregulated 33 genes coding for ribosomal proteins in RT2316-13. This was because a large number of ribosomes were needed only during rapid growth [27] and the slowed growth at low temperature promoted catabolic recycling of the older ribosomes to release amino acids and nucleotides. Nucleotides could be dephosphorylated, and hydrolyzed to ribose and nucleobases [28]. The action of ribokinase produces ribose-5-phosphate that can be converted to glycolytic intermediates (fructose-6-phosphate and glyceraldehyde-3-phosphate) in the nonoxidative pentose phosphate pathway.

The LT treatment downregulated four genes coding for enzymes in the citric acid cycle (fumarate hydratase, FC = −19.48; malate dehydrogenase, FC = −16.03; succinateCoA ligase, FC = −14.91; and aconitate hydratase, FC = −14.67), whereas a gene coding for isocitrate dehydrogenase [NADP+] (FC = 6.38) was upregulated. These results showed that the LT treatment depressed the activity of the TCA cycle, reducing energy supply and the production of citric acid. The decreased activity of the TCA cycle and NADPH-producing 6-phosphogluconate dehydrogenase explained the reduced lipid content (from 33.5% to 15.2% *w*/*w*) in RT2316-13 biomass grown at 5 °C [14].

Other similar observations have been reported. An earlier study with *Aurantiochytrium* found a reduced supply of biochemical energy from glycolysis and TCA cycle, as a consequence of cold stress (15 °C compared with 25 °C) [29]. The authors further observed that in the logarithmic growth phase, the expression of genes coding for a ribose 5-phosphate isomerase and ribose-phosphate pyrophosphokinase, the enzymes required for nucleotide synthesis, was upregulated. 

#### 2.2.4. Respiratory Chain and Oxidative Phosphorylation

Several genes of the respiratory chain were downregulated by the LT treatment. These were the genes coding for subunits of NADH dehydrogenase (ubiquinone) (3 genes) (FC = −23.18; −17.32; −15.98) (complex I), ATP synthase subunits (4 genes) (FC in the range of −17.36 to −15.37) (complex V), cytochrome c oxidase subunit (4 genes) (FC in the range of −26.45 to −15.40) (complex IV), and ADP/ATP carrier protein (catalyzes the exchange of ADP and ATP across the mitochondrial inner membrane) (FC = −19.26). The reduced activity of the respiratory chain and TCA cycle explained the reduced specific growth rate of RT2316-13 at the low temperature [14].

#### 2.2.5. Lipid Metabolism

Two main pathways are responsible of the synthesis of eicosapentaenoic acid (EPA) and docosahexaenoic acid (DHA). These are the polyketide synthase (PKS) system found in marine bacteria (*Shewanella* sp., [30]; *Vibrio* sp., [31]; *Photobacterium profundum*, [32]) and in some thraustochytrids (*Schizochytrium* sp., [33], *Thraustochytrium* sp. SZU445 [34]), and the elongase-desaturase system (ω3- and ω6-pathways) (shown in Figure 2). Precursors of ω6- and ω3-pathways (i.e. linoleic acid (LA, C18:2^Δ9,12^) and α-linolenic (ALA, C18:3^Δ9,12,15^), respectively), are synthetized from palmitic acid (PA, C16:0), the product of fatty acid synthase (FASN). Genes coding for FASN, Δ9- and Δ12-desaturases were found in the transcriptome of RT2316-13 (Table 3); however, the expression of a gene coding for Δ15-desaturase (ω3-desaturase), involved in ALA synthesis from LA, did not occur in RT2316-13 under the specified conditions. The trace ALA content (<1% *w*/*w* of total fatty acids) in RT2316-13 (Figure 2) could be explained by a bifunctional Δ12-desaturase as has been shown to occur in some fungi [35].

Genes coding for Δ4-, Δ6- and Δ8-desaturases, and elongases were expressed in RT2316-13 (Table 3). Although fatty acid elongation occurs through sequential action of four enzymes (condensation, reduction, dehydration, and reduction) adding two carbon units to the growing acyl chain, the rate and substrate specificity are governed by the first reaction, catalyzed by a fatty acid elongase. The elongases expressed by RT2316-13 were similar to the human elongases (ELOVL2, ELOVL4, and ELOVL6) (Table 3). The ω3- and ω6-pathways require Δ17- and Δ19-desaturases (ω3-desaturases) for the synthesis of DHA from the ω6 fatty acid substrates, but these enzymes were missing in RT2316-13. A key enzyme required for the synthesis of arachidonic acid (ARA, C20:4^Δ5,8,11,14^), EPA and DHA through the ω3- and ω6-pathways, is Δ5-desaturase. The transcript TRINITY_DN4828_c0_g2_i1 identified as a Δ8-fatty acid desaturase (SLD1_EUGGR, Table 3) was translated to protein, and queried by homology against non-redundant protein database in National Center for Biotechnology Information (NCBI) using the protein BLAST (Basic Local Alignment Search Tool) algorithm (https://blast.ncbi.nlm.nih.gov). The results showed a high identity (59.62%) and similarity (74.11%) with a Δ5-desaturase from *Oblongichytrium* sp. SEK 347 (accession BAG71007.1).

A homology search for genes in the PKS system involved in the EPA and/or DHA biosynthesis (*pfaA*, *pfaB*, *pfaC*, and *pfaD*) in the transcriptome of RT2316-13 was performed. The one significant hit after being extracted and queried by homology on GenBank and UniProt databases matched to a polyketide synthase (PKS) gene related to antibiotic biosynthesis. Thus, under the growth conditions, LCPUFA synthesis in RT2316-13 was not due to the PKS system, unlike in the related thraustochytrids *Schizochytrium* sp. [33] and *Thraustochytrium* sp. SZU445 [34].

Based on the enzymes identified in RT2316-13, the synthesis of EPA, DHA, and ARA starts from palmitic acid (Figure 2). The sequential action of ELOVL 6 (a fatty acid elongase), Δ9-desaturase and Δ12-desaturase produces stearic acid, oleic acid, and linoleic acid, whereas the action of a bifunctional Δ12/Δ15-desaturase on linoleic acid produces α-linolenic acid. The latter is transformed into eicosatrienoic acid, eicosatetraenoic acid, and EPA by the sequential action of Δ9-elongase, Δ8-desaturase, and Δ5-desaturase. The action of Δ5-elongase and Δ4-desaturase on EPA and DPA, respectively, produces DHA. The action of Δ9-elongase, Δ8-desaturase, and Δ5-desaturase on linoleic acid, eicosadienoic acid, and dihomo-γ-linolenic produces ARA.

Twenty-five genes coding for enzymes related to glycerolipid metabolism and biosynthesis of fatty acids and steroids were DE in RT2316-13. Among the DE genes coding for enzymes involved in the synthesis of PUFA were those coding for: a very-long-chain 3-oxoacyl-CoA reductase (FC = 8.32), a long-chain fatty acid CoA ligase (2 genes) (FC = 4.35; 6.08), ELOVL6 (FC = 6.41), ELOV2 (FC = 5.29), and a Δ8-fatty-acid desaturase (FC = 5.10). The latter gene was found to be highly similar to a Δ5-desaturase, a key enzyme in the synthesis of EPA and DHA. Upregulation of this gene together with upregulation of the gene coding for ELOV2 (a Δ5-elongase) explained the enrichment of the total lipid in EPA and DHA at LT (Figure 2).

Two genes coding for diacylglycerol O-acyltransferase 2, the enzyme required for the synthesis of triacylglycerols (TAG), were DE (FC = 14.62; −6.88), whereas two genes coding for monoacylglycerol lipase (MAGL) were upregulated (FC = 5.63; 12.24). MAGL catalyzes the last step in the hydrolysis of storage TAG and has been postulated to have a role in the detoxification of deleterious membrane lipids [38] which could form through oxidation of membrane PUFA because of the increased solubility of oxygen in water at LT [39,40].

In RT2316-13, a gene coding for cardiolipin synthase A was upregulated (FC = 4.60). Cardiolipins are phospholipids of the inner mitochondrial membrane required for the optimal activity of oxidative phosphorylation complexes and the ATP/ADP carrier [41]. Interactions of sterols with other membrane components contribute not only to regulate membrane fluidity and permeability, but also to the activity of membrane-bound enzymes and transporters [42]. Disruption of the SYRl/ERG3 gene that encoded sterol C-5 desaturase, an enzyme involved in the ergosterol biosynthesis, inhibited the growth of *S. cereviciae* at LT possibly because sterol was required for proper functioning of the tryptophan transporter at LT [43]. A similar function could explain the upregulation (FC = 20.09) of a Δ7-sterol-C5(6)-desaturase, involved in the biosynthesis of sitosterol and campesterol in RT2316-13. 

### 2.3. Validation of Gene Expression Profiles by qRT-PCR

The amplification of four selected genes that coded for acetyl-CoA carboxylase (AAC-5, AAC-15), malic enzyme (Me-5, Me-15), subunit A of PKS (Pfaa-5, Pfaa-15), and subunit C of LCPUFA PKS (Pfac-5, Pfac-15), in cDNA of the biomass grown at 5 °C and 15 °C was evaluated. The genes for AAC and Me were selected because the first of these is involved in the synthesis of fatty acid precursors, and the second one is involved in the synthesis of NADPH required for the synthesis of PUFA. Genes coding for acetyl-CoA carboxylase and malic enzyme were expressed at both the low and control temperatures (Appendix A). The lack of expression of the genes coding for the two subunits of LCPUFA PKS (Appendix A) was consistent with the results of transcriptome analysis. Five other genes coding for acetyl-CoA carboxylase and four other genes coding for malic enzyme were annotated in the transcriptome of RT2316-13, but these were not differentially expressed.

## 3. Materials and Methods 

### 3.1. Microorganism and Culture Conditions for RNA Extraction

The thraustochytrid *Oblongichytrium* sp. RT2316-13 used in this work had been isolated from seawater samples collected near the Antarctic Base Professor Julio Escudero (62°12′57″ S, 58°57′35″ E). Details of isolation and identification were previously published [15]. Briefly, the samples were supplemented with sterile pine pollen (~15 mg per 200 mL) and kept at 5 °C. After 15 days, pollen grains were collected by filtration (0.45 μm pore size) and dispersed on a solid medium supplemented with streptomycin sulfate and penicillin G (0.3 g/L of each) (Sigma, St. Louis, MO, USA) to prevent proliferation of bacteria. Pure isolates were obtained through repeated subculture of individual colonies on solid medium. Cell DNA was isolated and the sequence of the 18S rRNA gene was amplified using the method described previously [15]. Results of the Sanger sequencing were assembled using the Geneious 4.8.4 program (Biomatters Ltd., Auckland, New Zealand) and compared with sequences in EMBL/DDBJ/PDB/GenBank databases by BLASTN 2.2.21. Pure stock cultures were kept frozen at –80 °C in 50% glycerol (*v*/*v*).

Cells were grown aseptically. The culture medium comprised of glucose (20 gL^−1^; Merck, Darmstadt, Germany), yeast extract (10 gL^−1^; BBL™, Becton, Dickinson and Co., Sparks, MD, USA), and monosodium glutamate (0.6 gL^−1^; Merck) in a 1:1 by volume mixture of artificial sea water (ASW) [44] and distilled water. Trace elements and vitamins were added to the medium as previously specified [45]. Prior work had shown this medium to support growth with a highest attainable total lipid content in the biomass (35% of dry weight) at 15 °C [14]. 

Inoculum was prepared by transferring 1 mL of the thawed stock to 100 mL of the earlier specified liquid medium in a 250 mL Erlenmeyer flask. The flask was incubated (15 °C) on an orbital shaker at 150 rpm for 7 days. This culture (5 mL) was used to inoculate 100 mL of the same fresh medium and incubated (15 °C, 150 rpm) for 2 days. Afterwards, 50 mL aliquots of the grown culture were transferred to two flasks, each containing 50 mL of the same fresh medium. One of the flasks was incubated at 15 °C (control culture) and the other was incubated at 5 °C (low temperature (LT) culture) in an orbital shaker at 150 rpm. Both flasks were harvested after 3 days. 

Biomass was harvested by centrifugation (7000× *g*, 4 °C, 10 min). The recovered cell pellet was suspended in 12 mL of a 19:1 (by vol) mixture of aqueous phenol (0.8 M) and absolute ethanol, and stored at −20 °C until the extraction of the nucleic acids. Three replicate experiments were used to produce the control and LT biomass samples.

### 3.2. RNA Extraction

Cells from the previous step were lysed in a high-pressure homogenizer (Stansted Fluid Power Ltd., London, England) (one pass, 20 bar) under sterile conditions. Cell debris was removed from the lysate by centrifugation (10,000× *g*, 4 °C, 15 min). Total RNA was extracted using TRIzol reagent (Thermo Fisher Scientific Inc., Waltham, MA, USA) according to the manufacturer’s instructions. The collected RNA was purified using RNA purification columns (Thermo Fisher Scientific Inc., Waltham, MA, USA) following the manufacturer’s instructions.

Sample degradation and integrity were monitored by agarose gel electrophoresis (1% *w*/*v*, 100 mV, 30 min). The purity and recovery yield of RNA were calculated using the absorbance values measured at 260 and 280 nm using a UV spectrophotometer (BioTek Synergy HT Microplate Reader; BioTek Instruments, Inc., Winooski, VT, USA). 

### 3.3. Conversion of RNA to cDNA

The extracted RNA was treated with DNase to eliminate any contamination. One µL of random primers, mainly random hexanucleotides (500 µg mL^−1^; Promega Corporation, Madison, WI, USA) were added. A conversion mix (M-MLV RT 5×, M-MLV RT transcriptase, RNasin and dNTPs) was added. The first PCR program (MultiGene optiMAX; Labnet International Inc., Edison, NJ, USA) was: 5 min at 20 °C; 45 min at 37 °C; 5 min at 95 °C; and 4 °C until storage. A PCR using the primers for 18S rRNA subunit (Appendix A) was performed to verify the correct conversion. Conditions of the second PCR program were: 3 min at 95 °C; 30 cycles of 1 min each at 94 °C followed by 1 min at 61 °C, and 1 min at 72 °C; 10 min at 72 °C; and 4 °C until storage. The cDNA was stored for validation assays. 

### 3.4. Sequencing and Quality Control of RT2316-13 Transcriptome

A total of six samples were sequenced (Genoma Mayor, Santiago, Chile) in an Illumina HiSeq 4000 (Illumina, Inc., San Diego, CA, USA) platform for 150 cycles in paired-end mode. The sequencing outputs, in FastQ file format, were processed using TrimGalore software to remove adaptors and reads that were likely to be sequencing errors. The reads, which on average had a Phred score of less than 30 (1 error in 1000 bp), were removed from downstream analysis.

The remaining high quality reads, selected during the quality control step, were used to construct a *de novo* assembly of RT2316-13 transcriptome using the Trinity v2.9.1 software [46]. The software used de Bruijn graphs to reconstruct transcripts and their variants using an extensive K-mer search strategy. The assembly integrity was assessed using a set of conserved orthologous gene sequences contained in the Eukaryota OrthoDB v10 database (www.orthodb.org), which were queried by similarity to the assembled transcripts, as in the BUSCO approach (Benchmarking Universal Single-Copy Orthologs) described by Simão et al. [17].

### 3.5. Differential Gene Expression Analysis

The relative abundance was calculated using RSEM v1.2.26 [47]. The resulting abundance for each sample was merged in a matrix, and analyzed with the Bioconductor package edgeR in the R statistical environment [48]. The significance of changes in gene expression was judged using a False Discovery Rate value (FDR) of less than 0.05 and a minimum fold change (FC) of 4 as thresholds.

### 3.6. Transcriptome Annotation and Enrichment Analysis

All resulting genes were aligned into the UniProt/SwissProtKB database using BLAST+ with an e-value of 1e^−10^ as the threshold. Functional annotation and ontology assignments were performed using the PANTHER classification system [49] with gene lists obtained from blast results (top hit) aligned with the reference proteome database (version 2018_4) from the European Molecular Biology Laboratory (EMBL). To search for enrichment or depletion of an ontology term among differentially expressed genes, they were analyzed using GOSeq [50].

### 3.7. Validation of the Transcriptome Results

Conventional PCR was performed to confirm or discard the presence of genes associated with LCPUFA synthesis. Five pairs of primers were used: Subunits A and C of PKS, malic enzyme, acetyl-CoA carboxylase, and, as a housekeeping gene, the subunit 18S rRNA. The sequences were obtained from Ma et al. [29] (Appendix A). Green Master Mix (Promega, Madison, WI, USA) was used for PCR in accordance with the manufacturer’s instructions. The reaction was carried out in a final volume of 20 µL using 4 µL of cDNA synthesized as described above. Conditions of PCR program 2 (described above) were used. Results were verified by agarose gel (1% *w*/*v*) electrophoresis. 

### 3.8. Data Availability

The raw reads of the Sequence Read Archive (codes from SRR12600987 to SRR12600992) have been deposited at NCBI BioProject (accession number PRJNA642733). 

## 4. Conclusions

This study contributes to knowledge of the cold water marine thraustochytrid *Oblongichytrium* RT2316-13. This microorganism grows abundantly at 5 °C, a feature that is at least partly attributed to its ability to synthesize long chain omega-3 PUFA. Transcriptome analysis of *Oblongichytrium* RT2316-13 revealed a significant number of differentially expressed (DE) genes (2944) in response to a decrease in temperature. Most of the DE genes related to the composition of the cell membrane (fatty acids, sterols, phosphatidylinositol), the membrane-associated enzymes, and the membrane-linked cytoskeletal proteins and enzymes. This suggests that in this thraustochytrid, the cell membrane has a role in “sensing” temperature and is an active part of the temperature signal transduction system. 

Thraustochytrids are mainly studied because of their capability to synthesize docosahexaenoic acid (DHA). However, strains such as RT2316-13 produce lipids rich in both EPA and DHA, fatty acids with known health effects. An understanding of temperature modulation of the genes related to the biosynthesis of these fatty acids is important for designing efficient production processes. The present work shows that in RT2316-13 the synthesis of EPA and DHA occurs through the action of elongases and desaturases and thus differs from synthesis in *Aurantiochytrium* species. The upregulation of genes coding for a Δ5-desaturase and Δ5-elongase involved in the synthesis of EPA and DHA explained the enrichment of total lipid with these two long chain polyunsaturated fatty acids at the low temperature. 

## Figures and Tables

**Figure 1 marinedrugs-18-00563-f001:**
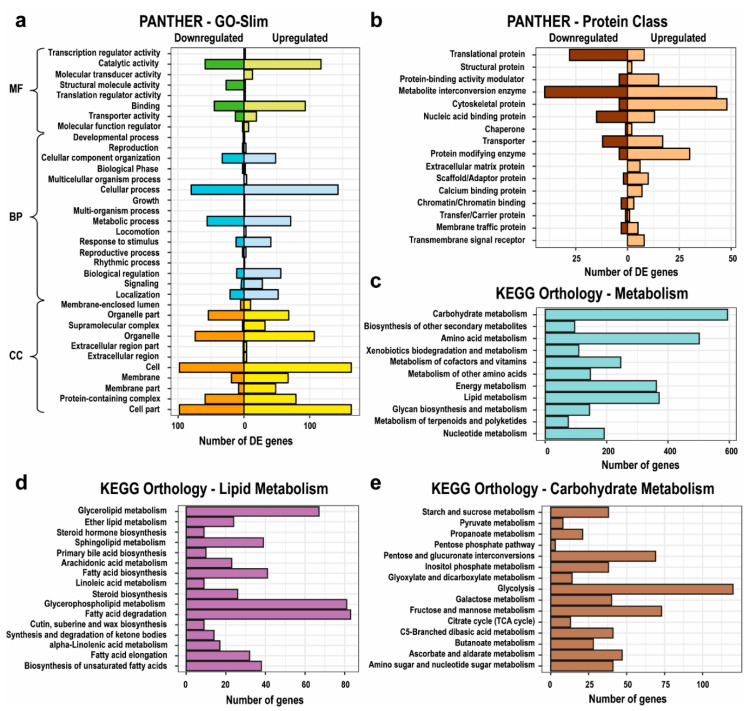
Number of differentially expressed (DE) genes in: (**a**) different GO-slim (cut-down versions of the Gene Ontologies) categories (molecular function, MF; biological process, BP; and cellular component, CC); and (**b**) different PANTHER protein classes. Distribution of the annotated genes in: (**c**) the different metabolisms; (**d**) lipid metabolism; and (**e**) carbohydrate metabolism, in the de novo assembled transcriptome of *Oblongichytrium* RT2316-13.

**Figure 2 marinedrugs-18-00563-f002:**
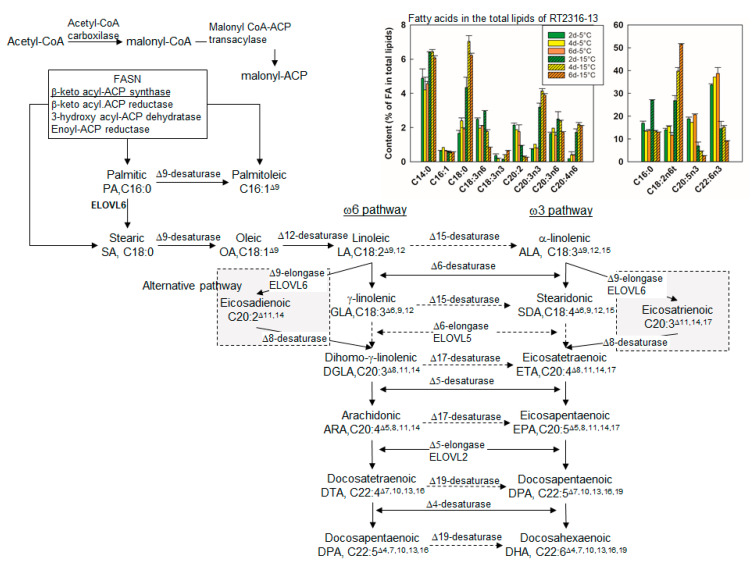
ω3- and ω6-pathways of biosynthesis of long chain polyunsaturated fatty acids and the alternative pathway (Δ6-elongase (ELOV6) + Δ8-desaturase). Reactions catalyzed by enzymes of the genes expressed in RT2316-13 are shown as continuous solid lines with arrowheads. Dashed lines with arrowheads denote reactions that would be catalyzed by enzymes/genes not expressed by RT2316-13 under the growth conditions. Inset graphs show the composition of the fatty acids in the total lipids of RT2316-13 at different temperatures (5 °C and 15 °C) and periods of incubation (2, 4 and 6 days) [11]. See reference [11] for further details.

**Table 1 marinedrugs-18-00563-t001:** Assembly metrics for the transcriptome of *Oblongichytrium* RT2316-13.

Metric	Value
Total raw reads (bp)	174,570,852
Total high quality reads (bp)	173,659,258
High quality content (%)	99.47
Number of transcripts	118,595
Number of genes	69,220
Total size (Mbp)	147.5
N50 (bp)	2518
Average length (bp)	1243
Median length (bp)	554

**Table 2 marinedrugs-18-00563-t002:** Summary of BUSCO ^§^
*de novo* transcriptome assembly of *Oblongichytrim* RT2316-13.

Library ^§^	Quantity	Percent of Total (%)
Complete BUSCOs	223	87.4
Single-copy BUSCOs	125	49.0
Duplicated BUSCOs	98	38.4
Fragmented BUSCOs	12	4.7
Missing BUSCOs	20	7.9
Total BUSCO genes	255	100

^§^ BUSCO (Benchmarking Universal Single-Copy Orthologs [17]).

**Table 3 marinedrugs-18-00563-t003:** Annotated genes in the transcriptome of *Oblongichytrium* RT2316-13 involved in the biosynthesis, elongation and desaturation of fatty acids.

Enzyme (Reaction)	EC Number	Swiss Prot ID
**Fatty acid biosynthesis**		
Acetyl-CoA carboxylase	6.4.1.2	ACACB_HUMAN; ACAC_YEAST; ACAC_SCHPO; ACACA_MOUSE
Fatty acid synthase, FASN	2.3.1.85	FAS_HUMAN
Fatty acid synthase subunit β	2.3.1.86	FAS1_CANAX
Fatty acid synthase subunit α	2.3.1.86	FAS2_PENPA
Fatty acid synthase β subunit aflB		ATNM_EMENI
Malonyl CoA-acyl carrier protein transacylase	2.3.1.39	FABD_BACSU
3-Oxoacyl-[acyl-carrier-protein] reductase FabG	1.1.1.100	FABG_THEMA; FABG_RICPR; FABG_VIBCH
3-Oxoacyl-[acyl-carrier-protein] synthase *	2.3.1.179	KASM_ARATH
Hydroxyacyl-thioester dehydratase type 2	4.2.1.	HTD2_HUMAN
Fatty acyl-CoA synthetase A	6.2.1.3	FCSA_DICDI
Very long-chain acyl-CoA synthetase	6.2.1.3	S27A2_MOUSE; S27A2_HUMAN
Enoyl-CoA hydratase ACTT6 ^§^	5.3.3.14	ACTT6_ALTAL
**Fatty acid elongation**		
3-Ketoacyl-CoA thiolase	2.3.1.16	THIKB_RAT
Trifunctional enzyme subunit β	2.3.1.16	ECHB_MACFA; ECHB_BOVIN
Trifunctional enzyme subunit α	2.3.1.16	ECHA_HUMAN; ECHA_PIG
3-Hydroxyacyl-CoA dehydrogenase	1.1.1.35	HCD2_DROME; HCDH2_CAEEL; HCDH1_CAEEL
Enoyl-CoA hydratase	4.2.1.17	ECHM_BOVIN; ECHM_DICDI; ECHM_RAT
Enoyl-[acyl-carrier-protein] reductase 1	1.3.1.; 1.3.1.38	ETR1_DEBHA; MECR_DICDI
Lysosomal thioesterase PPT2-A	3.1.2.22	PPT2A_XENLA
Palmitoyl-protein thioesterase 1	3.1.2.22	PPT1_MACFA
Elongation of very long chain fatty acids protein 2 (ELOVL2) ^¥^	2.3.1.199	ELOV2_HUMAN
Elongation of very long chain fatty acids protein 4 (ELOVL4) ^£^	2.3.1.199	ELOV4_HUMAN; ELOV4_MACMU
Elongation of very long chain fatty acids protein 6 (ELOVL6) ^‡^	2.3.1.199	ELO6_CAEEL; ELOV6_DANRE; ELOV6_MOUSE
Putative elongation of fatty acids protein	2.3.1.199	Y2012_DICDI
Very-long-chain 3-oxoacyl-CoA reductase	1.1.1.330	MKAR_LACBS; KCR1_ARATH
Very-long-chain (3R)-3-hydroxyacyl-CoA dehydratase	4.2.1.134	HACD_CAEEL
Very-long-chain enoyl-CoA reductase	1.3.1.93	TECR_ARATH; TECR_DICDI
Cytosolic acyl coenzyme A thioester hydrolase	3.1.2.2	BACH_HUMAN; BACH_RAT
**Biosynthesis of unsaturated fatty acids**		
Acyl-CoA desaturase (Δ9 desaturase)	1.14.19.1	ACOD2_MOUSE
Acyl-CoA desaturase 1	1.14.19.1	SCD1_TACFU
Acyl-CoA 6-desaturase	1.14.19.3	LLCD_SYNY3; FADS2_PONAB
Delta(12) fatty acid desaturase FAD2	1.14.19.6	FAD2_CALOF
Acyl-lipid (7-3)-desaturase (Δ4 desaturase)	1.14.19.31	D4FAD_EUGGR; D4FAD_THRSP
Sphingolipid delta(4)-desaturase	1.14.19.17; 1.14.18.5	DEGS_KOMPG; DEGS_CANAL
Delta(8)-fatty-acid desaturase	1.14.19.3	SLD1_EUGGR ^⁑^; SLD2_ARATH
Acyl-CoA thioesterase 2	3.1.2.	TESB_ECOLI

* Substrate specificity of this enzyme is similar to that of EC 2.3.1.41, but it differs in that palmitoleoyl-ACP is not a good substrate for it. This enzyme controls the temperature-dependent regulation of fatty-acid composition in *Escherichia coli* [36]. ^§^ The cis-3-enoyl product is required to form unsaturated fatty acids such as palmitoleic acid and cis-vaccenic acid, in dissociated (or type II) fatty-acid biosynthesis. ^¥^ Acts specifically on polyunsaturated acyl-CoA with a higher activity toward C20:4^Δ5,8,11,14^ and EPA-CoAs, among others [37]. Other substrates include DTA-CoA, EPA-CoA, DPA-CoA. ELOVL2 elongates. ^£^ ELOLV4 substrates: DTA-CoA, C26:4n6-CoA, C28:4n6-CoA, C30:4n6-CoA, C32:4n6-CoA, C34:4n6-CoA, C34:6n6-CoA, C24:0-CoA, C26:0-CoA, C28:0-CoA, C30:0-CoA, DHA-CoA, C24:5n3-CoA, C24:6n3-CoA, C26:5n3-CoA, C26:6n3-CoA, C28:5n3-CoA, C28:6n3-CoA, C30:5n3-CoA, C30:6n3-CoA, C32:5n3-CoA, C32:6n3-CoA, C34:5n3-CoA, C34:6n3-CoA, C36:5n3-CoA. ^‡^ ELOLV6 substrates: C12:0-CoA, C14:0-CoA, C16:0-CoA, C16:1^Δ9^-CoA, C18:1^Δ9^-CoA, C18:2^Δ9,12^-CoA, C18:3^Δ9,12,15^-CoA. ^⁑^ Transcript TRINITY_DN4828_c0_g2_i1 was translated to protein, and queried by homology against non-redundant protein database in National Center for Biotechnology Information (NCBI) using the protein BLAST (Basic Local Alignment Search Tool) algorithm (https://blast.ncbi.nlm.nih.gov). The results showed a high identity (59.62%) and similarity (74.11%) with a Δ5-desaturase of *Oblongichytrium* sp. SEK 347 (accession BAG71007.1).

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
