# Peer review of "Temperature Differentially Affects Gene Expression in Antarctic Thraustochytrid Oblongichytrium sp. RT2316-13"

_marinedrugs, 2020, doi:10.3390/md18110563_

Round 1

Reviewer 1 Report

In the paper “Temperature differentially affects gene expression in 3 Antarctic thraustochytrid Oblongichytrium sp. 4 RT2316-13. Authors have described the molecular mechanism exploited for the production of microbial lipids with a fatty acid profile. Marine thraustochytrids are good producers of lipids rich in docosahexaenoic acid (DHA), which are also rich source of human food sources. Therefore profiling of lipids is much needed in these resources.

This work is ready to publish after minor revision. However, I have few comments and I recommend the authors to address. My comments are below.

  • “The upregulation of the genes coding for a D5-desaturase and a D5-elongase involved in the synthesis of EPA and DHA, explained the enrichment of total lipid with these two long-chain fatty acids at the low temperature.” How did author determine the concentration of EPA and DHA?
  • Briefly explain the method of isolation and identification of the thraustochytrid in the respective section.
  • How lipids were isolated from biomass? Explain in method section separately.
  • What technique was used to measure EPA and DHA? GC-MS or other?
  • Suggestion for future studies. 13C-labeled tracer (e.g. [U-13C]glucose) can be used for the enrichment of fatty acids in different condition. Lipid flux can be measured by using 13C-labeled tracer which will be true flux measured in given time of exposure of tracer. 

Author Response

Responses to Reviewer 1' comments

Comment 1. “The upregulation of the genes coding for a D5-desaturase and a D5-elongase involved in the synthesis of EPA and DHA, explained the enrichment of total lipid with these two long-chain fatty acids at the low temperature.” How did author determine the concentration of EPA and DHA?

Response 1. The total lipids in the freeze-dried biomass were extracted and methylated as described in our earlier work cited in the manuscript (references 18 and 19). Fatty acid composition of the methylated lipid extract was determined using a gas chromatograph (reference 51, cited in the manuscript). As the data for total lipids in Figure 2 came from our previously published work (reference 11), as noted clearly in the caption of the figure (line 249), the measurement methods did not need explaining in the present paper. The original reference should be consulted for these. This has been clarified in the revised text (line 249, highlighted text).

Comment 2. Briefly explain the method of isolation and identification of the thraustochytrid in the respective section.

Response 2. A brief description has been provided (lines 316-324), as requested.

Comment 3. How lipids were isolated from biomass? Explain in method section separately. What technique was used to measure EPA and DHA? GC-MS or other?

Response 3. Lipid extraction and analysis data came from our previously published paper, as emphasized in the caption of Figure 2 (line 249). The original reference should be consulted for the relevant details, as further clarified in the revised text (line 249).

Comment 4. Suggestion for future studies. C-labeled tracer (e.g. [U- C]glucose) can be used for the enrichment of fatty acids in different condition. Lipid flux can be measured by using C-labeled tracer which will be true flux measured in given time of exposure of tracer.

Response 4. We appreciate the reviewer’s suggestions for future studies.

Reviewer 2 Report

This is a well-written and presented manuscript and seems to be a revision; however the original manuscript was also attached. If space allows I would suggest to expand on the rationale and most importantly compare with other like studies if applicable or also make statements as to the novelty of the study.

Author Response

Responses to Reviewer 2' comments

Comment 1. This is a well-written and presented manuscript and seems to be a revision; however the original manuscript was also attached. If space allows I would suggest to expand on the rationale and most importantly compare with other like studies if applicable or also make statements as to the novelty of the study.

Response 2. The “novelty” is in a clear delineation of the differential gene expression in response to a step change in temperature. No such information has ever been published for the thraustochytrid of interest in the present work. Significantly, based on the gene expression studies, the mechanism of DHA synthesis in the thraustochytrid of interest is shown to differ from species of the genera such as Aurantiochytrium. This has been noted in the revised text (line 417-419).

Reviewer 3 Report

Oblongichytrium sp. RT2316-13 isolated from Antarctic seawaters accumulates lipids rich in DHA and EPA. The authors investigated the gene expression profile of Oblongichytrium sp. RT2316-13 cultured at different temperatures. The genes whose expression levels change depending on the culture temperature were revealed, and these results could explain the characteristic phenotypes of Oblongichytrium sp. RT2316-13 under low temperature environment. My evaluation is that this manuscript is publishable with minor corrections. My comments about this paper is as follows.

MINOR COMMENTS

L57

The description, "most of the studied thraustochytrids are able to produce DHA as the single long chain polyunsaturated fatty acid (LCPUFA) " should be rethought. The well-studied members of thraustochytrid such as Aurantiochytrium and Schizochytrium accumulate not only DHA but also DPA.

L134

What are the five protein classes left other than cytoskeleton, metabolite interconversion and protein modification? Please indicate all the protein classes with significant change for reader-friendly description.

L193

The abbreviation, "PLC" is not defined in the revised text.

L265 (Lipid metabolism)

It is difficult to understand the synthetic pathways of EPA and DHA in Oblongichytrium sp. RT2316-13. The authors should clarify the synthetic pathway of PUFA in the absence of PUFA synthase, ∆17 desaturase, and ∆19 desaturase.

L358

This result can explain the absence of the gene expression of PUFA synthase in Oblongichytrium sp. RT2316-13, however the reason why the genes of AAC and Me were selected as targets was not described. This point should be clarified.

Figure 1e

This figure is not referred in the text. Please refer it at the appropriate part.

Author Response

Responses to Reviewer 3' comments

Comment 1. L57. The description, "most of the studied thraustochytrids are able to produce DHA as the single long chain polyunsaturated fatty acid (LCPUFA)" should be rethought. The well-studied members of thraustochytrid such as Aurantiochytrium and Schizochytrium accumulate not only DHA but also DPA.

Response 1. The text in question (line 51) has been revised to address this issue. DHA is typically the main long-chain polyunsaturated fatty acid.

Comment 2. L134. What are the five protein classes left other than cytoskeleton, metabolite interconversion and protein modification? Please indicate all the protein classes with significant change for reader-friendly description.

Response 2. The relevant protein classes have been clarified in the revised text (lines 112-117), as requested.

Comment 3. L193. The abbreviation, "PLC" is not defined in the revised text.

Response 3. The abbreviation in question has been explained in the revised text (line 164), as requested.

Comment 4. L265 (Lipid metabolism). It is difficult to understand the synthetic pathways of EPA and DHA in Oblongichytrium sp.RT2316-13. The authors should clarify the synthetic pathway of PUFA in the absence of PUFAsynthase, Δ17 desaturase, and Δ19 desaturase.

Response 4. The explanation of the synthetic pathway has been improved (lines 269-276), as requested.

Comment 5. L358. This result can explain the absence of the gene expression of PUFA synthase in Oblongichytrium sp. RT2316-13, however the reason why the genes of AAC and Me were selected as targets was not described. This point should be clarified.

Response 5. The choice of AAC and Me has been clarified (lines 303-305), as requested.

Comment 6. Figure 1e. This figure is not referred in the text. Please refer it at the appropriate part.

Response 6. Figure 1e has been cited in the revised text (lines 122-124), as requested.